# TORCH GEOMETRIC POOL: THE PYTORCH LIBRARY FOR POOLING IN GRAPH NEURAL NETWORKS

## ABSTRACT

We introduce Torch Geometric Pool (`tgp`), a library for hierarchical pooling in Graph Neural Networks. Built upon Pytorch Geometric, Torch Geometric Pool (`tgp`) provides a wide variety of pooling operators, unified under a consistent API and a modular design. The library emphasizes usability and extensibility, and includes features like precomputed pooling, which significantly accelerate training for a class of operators. In this paper, we present `tgp`'s structure and present an extensive benchmark. The latter showcases the library's features and systematically compares the performance of the implemented graph-pooling methods in different downstream tasks. The results, showing that the choice of the optimal pooling operator depends on tasks and data at hand, support the need for a library that enables fast prototyping.

## 1 INTRODUCTION

Graph Neural Networks (GNNs) (Gori et al., 2005; Scarselli et al., 2008; Bronstein et al., 2021) have become a cornerstone of machine learning for structured data. Inspired by the success of hierarchical feature learning in Convolutional Neural Networks, graph-pooling operations have been introduced to enable GNNs to learn multi-scale representations (Wang et al., 2024). By mapping a graph to a coarser version of itself, graph-pooling operators – hereinafter named *poolers* – can capture high-level structural information, expand the receptive field of a GNN, and even reduce computational complexity (Gal et al., 2025). Despite the importance of pooling, its practical application has been hampered by a lack of unified software implementations. Popular GNN frameworks like Pytorch Geometric (PyG) (Fey & Lenssen, 2019) and Spektral (Grattarola & Alippi, 2021) provide only a few of the oldest pooling operators. In the former, pooling is not standardized under a unified API, while the latter belongs to the Tensorflow ecosystem, which nowadays is rarely used in GNN research. Finally, Deep Graph Library (Wang et al., 2019) only focuses on the implementation of Message Passing (MP) and global pooling operators. This makes it difficult for researchers and practitioners to directly compare different poolers and rapidly prototype new hierarchical GNNs.

To resolve these challenges, we introduce `tgp`, a dedicated library that provides a comprehensive, easy-to-use, and extensible toolkit for hierarchical graph pooling. Built as a seamless extension to PyG, its purpose is to consolidate the vast landscape of pooling methods into a single, coherent framework. By doing so, it greatly facilitates building and evaluating hierarchical GNNs. The main features of the library are summarized as follows:

- **Unified and consistent API.** Our primary contribution is a library that implements a wide variety of graph pooling operators under a unified API, allowing for exchanging poolers within a GNN with a single line of code. This consistency is crucial for eliminating ad-hoc implementations and enabling fair and systematic comparisons.

- **Modular and extensible design.** `tgp` is inspired by the modular Select-Reduce-Connect-Lift (SRC(L)) framework. This design promotes extensibility by breaking down each pooling operator into a few fundamental components. Consequently, users can easily create novel pooling operators by combining new and existing modules. This lowers the barrier to prototyping and innovation both in research and industry.

- **Caching and pre-coarsening.** We introduce caching and pre-coarsening mechanisms that significantly speed up the training of deterministic poolers. Such operators often rely on graph-theoretic

properties that are expensive to compute at every forward pass. Through caching and pre-coarsening, pooling is performed only once; the results, which can be saved on the disk, are reused across mini-batches, dramatically reducing training times and enabling applications on large-scale graphs.

To showcase the library's functionalities, we present an extensive benchmarking of existing poolers on both graph-level and node-level tasks, which systematically evaluates the performance and trade-offs of different pooling families. Our findings indicate that choosing the pooler is a critical, task-dependent decision, which makes `tgp` an essential tool to facilitate this exploration.

## 2 BACKGROUND AND MOTIVATION

### 2.1 SRC(L): A UNIFIED GRAPH POOLING FRAMEWORK

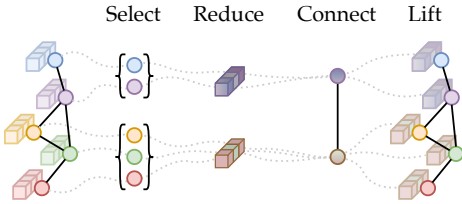

Figure 1: Overview of SRC(L). The SRC stages coarsen the graph by mapping nodes to supernodes. Lifting is the inverse operation that reprojects supernodes back to the original node space.

While there are profound differences between existing graph pooling operators, most of them can be expressed through the SRC(L) framework (Grattarola et al., 2022). Specifically, a pooling operator $\text{POOL}: (\boldsymbol{A}, \boldsymbol{X}) \mapsto (\boldsymbol{A}', \boldsymbol{X}')$ can be expressed as the combination of the following sub-operators, as illustrated in Fig. 1:

- **Select**: $(\boldsymbol{A}, \boldsymbol{X}) \mapsto \boldsymbol{S} \in \mathbb{R}^{N \times K}$, defines how the $N$ original nodes are mapped to the $K$ pooled nodes, called *supernodes*. The output $\boldsymbol{S}$ is the *selection matrix* (or assignment matrix).

- **Reduce**: $(\boldsymbol{X}, \boldsymbol{S}) \mapsto \boldsymbol{X}' \in \mathbb{R}^{K \times F}$, yields the features of the supernodes based on the original features and the selection matrix. For example, $\boldsymbol{X}' = \boldsymbol{S}^\top \boldsymbol{X}$.

- **Connect**: $(\boldsymbol{A}, \boldsymbol{S}) \mapsto \boldsymbol{A}' \in \mathbb{R}_{\geq 0}^{K \times K}$, generates the new adjacency matrix for the coarsened graph, by merging the edges of the nodes mapped to the same supernode, e.g., $\boldsymbol{A}' = \boldsymbol{S}^\top \boldsymbol{A} \boldsymbol{S}$.

Furthermore, to support node-level tasks, this framework is often complemented by a fourth operation:

- **Lift**: $(\boldsymbol{X}', \boldsymbol{S}) \mapsto \boldsymbol{X}_{\text{lift}} \in \mathbb{R}^{N \times F'}$, projects the features of the supernodes back to the original $N$ nodes, e.g., $\boldsymbol{X}_{\text{lift}} = \boldsymbol{S} \boldsymbol{X}'$.

A typical workflow for a node-level task is: i) apply one or more MP to the node original node features $\boldsymbol{X}$; ii) perform pooling to obtain a coarsened graph $\{\boldsymbol{A}', \boldsymbol{X}'\}$; iii) apply MP to $\boldsymbol{X}'$; iv) lift the coarsened node features back to the original node space; v) apply a readout to $\boldsymbol{X}_{\text{lift}}$ to compute the output for each node. See Appendix D.2 for a more detailed example.

### 2.2 A TAXONOMY OF GRAPH POOLING METHODS

Similar to Grattarola et al. (2022), we categorize graph pooling methods along two primary, orthogonal axes that highlight key implementation challenges: the presence of learnable parameters (**trainable vs. non-trainable**) and the structure of the selection matrix (**sparse vs. dense**). Fig. 2 illustrates the differences among different types of pooling operators.

**Trainable vs. non-trainable**   A fundamental distinction among pooling operators is whether the coarsened graph is obtained either through trainable or pre-computed operations. *Non-trainable methods* usually rely on graph-theoretic algorithms that account only for the graph's topology. This implies that the pooled graph's topology does not depend on the node features and the downstream

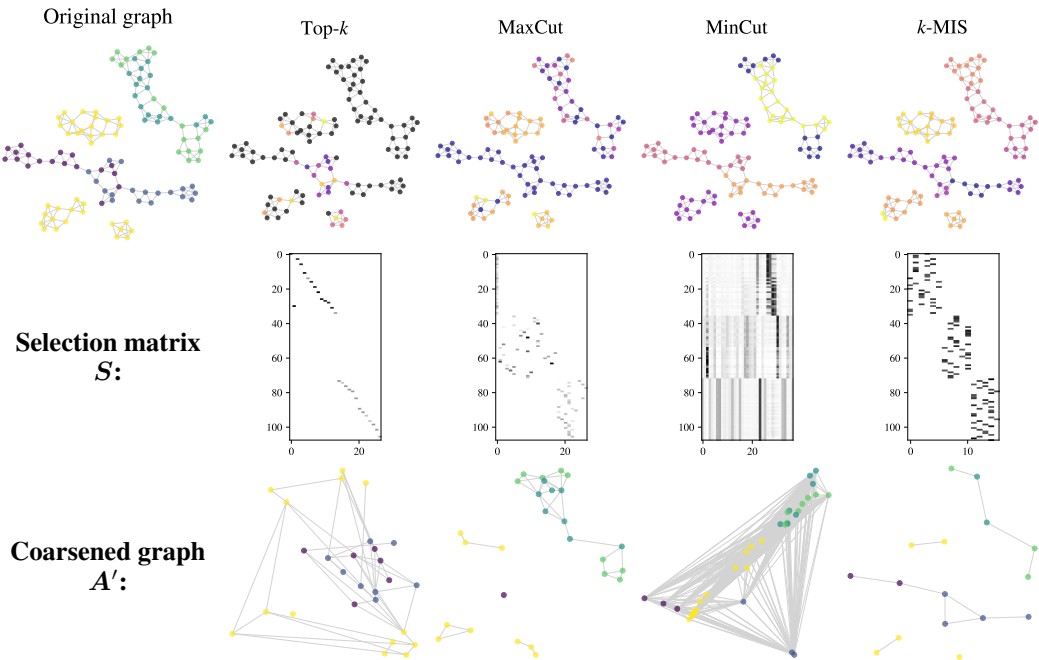

Figure 2: Pooling examples with different operators. **1ˢᵗ row**: original graph with color-coded features and node-to-supernodes assignments by different operators; Top-$k$ (sparse, trainable) selects non-black nodes based on the score (color); MaxCut (sparse, trainable), MinCut (dense, trainable), and $k$-MIS (sparse, non-trainable) assign nodes with the same color to one supernode. **2ⁿᵈ row**: the cluster assignment matrix $S$ of each operator. **3ʳᵈ row**: the pooled graphs with node features color-coded as in the original graph.

task. Examples include methods based on spectral clustering (Dhillon et al., 2007), largest eigenvector vertex selection (Bianchi et al., 2020b), and maximal independent sets (Bacciu et al., 2023). Since the coarsening is feature-independent, the pooled graphs can be pre-computed, simplifying training at the cost of expressivity, as the pooling strategy does not adapt to the task at hand. *Trainable methods*, instead, incorporate learnable parameters that are optimized end-to-end with the rest of the GNN. A trainable pooler can identify the most important nodes or substructures for the task at hand, often leading to superior performance at the cost of computational complexity.

**Sparse vs. dense assignment** Pooling operators can be further divided based on the structure of the assignment matrix $S$ they produce. This distinction significantly affects memory footprint, computational complexity, and the structure of the pooled graph. In *Sparse poolers*, the assignment matrix $S \in \{0, 1\}^{N \times K}$ has a number of non-zero entries proportional either to the number of supernodes ($K$) or to the number of nodes in the original graph ($N$). Methods such as Top-$k$ (Gao & Ji, 2019; Knyazev et al., 2019) learn a scoring function to determine which nodes to select. This approach is highly efficient but leads to information loss, as non-selected nodes are discarded entirely. *Dense poolers*, often called *soft-clustering* poolers, produce a dense assignment matrix $S$, which gives the $N$ original nodes a "soft" membership to each of the $K$ supernodes (clusters). Representatives like Diffpool (Ying et al., 2018) and MinCut (Bianchi et al., 2020a) learn this differentiable node partitioning guided by auxiliary losses. While highly expressive (Bianchi & Lachi, 2023), these poolers are computationally demanding and typically require specifying a fixed number of clusters $K$, which is problematic for datasets with graphs of variable sizes.

## 3 THE TORCH GEOMETRIC POOL LIBRARY

To address the fragmentation in pooling methods and in their implementation, we designed tgp based on SRC(L) framework, which provides a principled and consistent framework. The core design goals are usability, modularity, and efficiency, which we achieve by leveraging the taxonomy of

the pooling methods. `tgp` provides a comprehensive collection of operators that can be seamlessly integrated within a GNN implemented with PyG. Listing 1 provides a conceptual example of how these components are used to achieve this flexibility in a typical GNN made with PyG. A table summarizing the features of each pooler is available in Appendix A

```python
class GNN(torch.nn.Module):
    def __init__(self, pooler_type, **pooler_kwargs):
        ...
        # Instantiate any pooler from tgp
        self.pooler = get_pooler(pooler_type, **pooler_kwargs)

        # Conditional model construction
        if self.pooler.is_dense:
            self.conv2 = DenseGCNConv(...)
        else:
            self.conv2 = GCNConv(...)

        self.readout = MLP(...)

    def forward(self, data, supervision_loss_fn):
        # ... pre-pooling layers ...

        # Conditional data processing and pre-coarsening
        if self.pooler.is_precoarsenable and data.num_graphs == 1:
            caching = True
        x, adj, ... = self.pooler.preprocessing(..., use_cache=caching)

        out = self.pooler(x=x, adj=adj, ...)

        # ... post-pooling layers using the conditional conv2 ...

        # Unified readout
        graph_embedding = self.pooler.global_pool(out.x, batch=out.batch)
        predictions = self.readout(graph_embedding)

        # Conditionally add auxiliary loss to the main objective
        loss = supervision_loss_fn(predictions, data.y)
        if self.pooler.has_loss:
            loss += out.loss

        return loss
```

Listing 1: Demonstration of the unified API. The `preprocessing` call standardizes the input, attributes like is_dense, is_precoarsenable, and has_loss condition the model's architecture and the loss calculation, and the `global_pool` method provides a consistent readout.

## 3.1 A UNIFIED API

Sparse and dense poolers produce outputs with fundamentally different structures, which hinders the interchangeability of these methods within the same GNN architecture. Differences span from the assignment matrix's structure to how batches of graphs are handled: sparse methods use PyG's disjoint union approach, while dense methods require padded tensors with an explicit batch dimension. `tgp` provides a single root class – namely `SRCPooling` – across pooling families and reconciles the discrepancies by offering a common API that streamlines the entire pooling workflow through standardized mini-batching, conditional model construction, and a unified readout operation.

**Standardized mini-batching.** Every pooler comes with a `preprocessing` method which ensures that the data passed to the pooler's `forward` call is always in the correct format. In dense poolers, a mini-batch of graphs is created by padding and stacking the graphs in the batch, resulting in a tensor of shape $[B, N, N]$, with $B$ being the number of graphs and $N$ the number of nodes in the largest graph. A binary mask is also generated to identify the padded nodes. In sparse poolers, instead, `preprocessing` is typically a pass-through.

**Conditional model construction.** To account for different requirements in data-handling and training modalities, poolers expose three class properties that ease conditional selections:

- `is_dense`: Checks if the pooler outputs a dense graph, allowing for the conditional selection of the appropriate GNN layer (e.g., `GCNConv` vs. `DenseGCNConv`).

- `has_loss`: Indicates whether the pooler provides an auxiliary loss, which can be either combined with other training losses or optimized alone in a self-supervised task.

- `is_precoarsenable`: Indicates if a pooler is non-trainable and if its Select and Connect operations can be cached or pre-computed.

**Unified readout.** The final step in graph-level tasks is typically a *global pooling* (readout) operation that produces a single embedding for each graph. This operation is implemented differently for sparse (using a `batch` indicator vector) and dense (reducing across the node dimension) methods. `tgp` abstracts this away by providing for each pooler a `global_pool()` method, which has a consistent signature and internally dispatches to the correct implementation. While global pooling in `tgp` aggregates the node features through simple reductions (e.g., sum or mean), more sophisticated operations are possible by calling any `Aggregator` implemented in PyG[1].

### 3.2 MODULARITY AND EXTENSIBILITY

The SRC(L) framework's decomposition of pooling into distinct stages naturally provides a modular software design where the Select, Reduce, Connect, and Lift stages are implemented as interchangeable components. This architecture is the backbone of `tgp`'s extensibility. To make this concrete, in Listing 2 we sketch the implementation of the Top-$k$ operator, which is constructed by combining specific SRC(L) components in its `__init__` method. The `forward` pass then orchestrates the whole pooling logic through a sequence of calls to these components.

```python
class TopkPooling(SRCPooling):
  def __init__(self, in_channels, ratio, **kwargs):
    super().__init__(
        selector=TopkSelect(in_channels=in_channels, ratio=ratio, ...),
        reducer=BaseReduce(),
        connector=SparseConnect(),
        lifter=BaseLift())

  def forward(self, x, edge_index, batch=None):
    # 1. Select nodes via the TopkSelect component
    select_out = self.select(x=x, batch=batch)

    # 2. Reduce features via the BaseReduce component
    x_pooled, batch_pooled = self.reduce(x=x, so=select_out, batch=batch)

    # 3. Connect graph via the SparseConnect component
    edge_index_pooled, _ = self.connect(so=select_out, edge_index=edge_index)

    # 4. Return standardized output
    return PoolingOutput(x=x_pooled, edge_index=edge_index_pooled,
        batch=batch_pooled, so=select_out)
```

Listing 2: Sketched implementation of `TopkPooling`, showcasing the modular design. The `__init__` defines the SRC(L) components of the pooler, which are then called in the `forward` pass.

This modularity is achieved through standardized data structures that serve as interfaces between the components. The `select` method returns a `SelectOutput` object, which contains the assignment matrix $S$ and provides powerful utility methods (e.g., `is_sparse`, `is_expressive`, etc.). Similarly, the final output of any pooler is a `PoolingOutput` object, which consistently bundles the coarsened graph, batch information, and any auxiliary losses. This modular design, which decouples components via these data structures (further discussed in Appendix B), reduces boilerplate code, facilitating prototyping and experimentation. For example, to experiment with a new node scoring mechanism,

---

[1]https://pytorch-geometric.readthedocs.io/en/latest/modules/nn.html#aggregation-operators

one could simply write a new `Select` module and plug it into the `TopkPooling` class, reusing the existing components that perform Reduce and Connect operations. Similarly, one could instantiate hybrid poolers by replacing the standard sparse connector with a `KronConnect` module. Listing 3 illustrates how to swap the connector of a Top-$k$ pooler with a single line of code.

```python
from tgp.poolers import get_pooler
from tgp.connect import KronConnect

# 1. Instantiate a standard TopK pooler
pooler = get_pooler("topk", in_channels=64, ratio=0.5)

# 2. Hot-swap the connector component
# We replace standard connect operation with a Kron reduction based one
pooler.connector = KronConnect()
```

Listing 3: Instantiating a custom pooling layer by overriding the connector component.

### 3.3 EFFICIENCY IMPROVEMENTS VIA CACHING AND PRE-COARSENING

In non-trainable pooling methods, the assignment matrix $S$ is determined by a fixed, feature-independent algorithm, which can be expensive and wasteful to recompute at every forward pass during training. `tgp` offers two distinct and complementary optimization mechanisms: in-memory caching and a pre-coarsening data transform.

**In-memory caching for single-graph datasets.** For tasks that operate on a single, static graph (e.g., transductive node classification), we provide a simple caching mechanism. By setting the `cached=True` flag in any non-trainable pooler's constructor, the results of the expensive Select and Connect operations are computed just once on the first forward pass and stored internally within the pooler object. Every subsequent call bypasses these steps and performs only the Reduce operation (typically a single matrix multiplication). This process is completely transparent to the user and significantly speeds up training.

**Pre-coarsening for multi-graph datasets.** The in-memory caching approach is not suitable for tasks like graph classification, where each mini-batch contains graphs of varying sizes. For this (common) scenario, we provide a more powerful pre-coarsening mechanism, implemented as a standard PyG `pre_transform`, which is applied to the dataset before training. The transform iterates through every graph, applies the non-trainable pooler's Select and Connect operations, and saves the resulting assignment matrix $S$ and coarsened adjacency matrix $A'$ within the original `Data` object. To correctly form mini-batches of these `Data` objects, we provide a custom `PoolDataLoader`. This loader implements a custom `collate` function, which correctly joins the pre-coarsened structures from each graph in the batch into a single `PooledBatch` object, making the pre-computed information readily accessible to the model. During the training loop, the GNN can access the `PooledBatch`, bypassing the expensive pooling calculations and performing only the fast Reduce step. The logic in the `forward` pass becomes a simple check:

```python
def forward(self, data):
    # ... pre-pooling layers on original graph ...

    if hasattr(data, "pooled_data"):
        # Precoarsened data exists: only perform the RED step
        x_pooled, _ = self.reducer(x=x, so=data.pooled_data.so)
        adj_pooled = data.pooled_data.edge_index
    else:
        # Standard path: perform full SEL, RED, CON
        out = self.pooler(x, data.edge_index, ...)
        x_pooled, adj_pooled = out.x, out.edge_index

    # ... post-pooling layers on the coarsened graph ...
```

Listing 4: Conceptual forward pass using pre-computed data to bypass expensive pooling operations.

Since the pre-coarsening transform occurs offline and results are stored on disk, it circumvents the GPU memory constraints of processing large graphs at runtime. During training, the system simply loads the pre-computed pooled graph alongside the original input, avoiding the heavy computational and memory overhead of on-the-fly execution.

Both caching and pre-coarsening yield a dramatic reduction in training time, which is particularly impactful for large graphs or when training for many epochs, significantly accelerating the operational pipeline. The unified API of `tgp` allows for fully leveraging the computational advantages of non-trainable methods, while maintaining the same interface as trainable ones.

## 4    GRAPH POOLING BENCHMARK

We conduct an extensive experimental evaluation to showcase the capabilities of `tgp` and to systematically compare the performance of different pooling operators. Our experiments focused on the trade-offs between predictive performance, computational cost, and task suitability. Our evaluation is divided into four parts: i) unsupervised node clustering; ii) node classification; iii) graph classification; iv) efficiency analysis. We designed the experiments so that each pooler appears in at least one of the tasks where it is relevant, and we compare representatives from each family (sparse or dense, trainable or not) in the more general graph classification task. For each pooler, we use the default hyperparameter configuration provided by `tgp`. Notably, conducting such a broad benchmark involving operators with fundamentally different architectures and training requirements was made straightforward by the unified and modular API of `tgp`.

### 4.1    UNSUPERVISED NODE CLUSTERING

This experiment evaluates the ability of dense pooling operators to identify, within a single graph and in a purely unsupervised fashion, communities that align well with the node labels. We train a GNN to generate a node partition by optimizing only the auxiliary losses provided by assessed pooling layers, without any supervised signal. The partitions are given by the assignment matrix $S$ of six dense pooling operators: Asymmetric Cheeger Cut pooling (ACC) (Hansen & Bianchi, 2023), Diffpool, Deep Modularity Networks (DMoN) (Tsitsulin et al., 2023), High-Order Spectral Clustering (HOSC) (Duval & Malliaros, 2022), Just-Balance Pooling (JBPool) (Bianchi, 2022), and MinCut. The detailed experimental protocol, including the GNN architecture and datasets, is described in Appendices C.1 and D.1, respectively. We note that changing the hyperparameters in an unsupervised task like this one can significantly modify the outcome. However, tuning the hyperparameters in a clustering setting without relying on supervised information is not straightforward and out of scope for this evaluation.

Table 1: NMI by different poolers on node clustering. For each dataset: best performing pooler (**bold**); second best (underlined). Score: sum of points (2 for best, 1 for second best) across datasets.

| Pooler | CiteSeer | Community | Cora | DBLP | PubMed | Score |
|---|---|---|---|---|---|---|
| ACC | **27** $\pm 3$ | 91 $\pm 2$ | **41** $\pm 3$ | 26 $\pm 4$ | **22** $\pm 4$ | **6** |
| Diffpool | 16 $\pm 2$ | 50 $\pm 2$ | 29 $\pm 2$ | 15 $\pm 1$ | 10 $\pm 4$ | 0 |
| DMoN | 22 $\pm 1$ | **97** $\pm 0$ | 36 $\pm 3$ | 28 $\pm 2$ | 19 $\pm 8$ | 4 |
| HOSC | 20 $\pm 5$ | 83 $\pm 8$ | 29 $\pm 3$ | **34** $\pm 0$ | 17 $\pm 6$ | 2 |
| JBPool | 19 $\pm 4$ | 94 $\pm 8$ | 22 $\pm 6$ | 22 $\pm 8$ | 16 $\pm 4$ | 1 |
| MinCut | 21 $\pm 3$ | 89 $\pm 1$ | 39 $\pm 3$ | 31 $\pm 6$ | 18 $\pm 3$ | 2 |

Table 1 shows that operators with strong graph-theoretic inductive biases, such as ACC and DMoN, excel at creating node partitions that align with the true class of the nodes. Optimizing well-established graph partitioning objectives, such as the minimum cut and the graph modularity, works particularly well in homophilic graphs where the nodes of a community have similar class labels. In contrast, Diffpool consistently underperforms in terms of Normalized Mutual Information (NMI) as it optimizes losses inspired by heuristics rather than graph-theoretical objectives: a link prediction loss that encourages connected nodes to be in the same cluster and an entropy term that prevents cluster assignments from being too smooth. Without the primary learning signal provided by a supervised

objective, these regularizers alone are often insufficient to guide the model toward learning meaningful partitions. In summary, the results underscore that in an unsupervised tasks the result depends heavily on the alignment between the objectives optimized by its auxiliary losses, the structure of the graph, and the properties of its nodes. Results with additional clustering metrics are reported in Appendix E.

## 4.2 NODE CLASSIFICATION

This experiment evaluates the performance of various pooling operators in transductive node classification on a heterophilic setting. The task is particularly challenging as it requires the model to first pool the graph and then project the learned information back to the original node space for prediction using the `Lift` (un-pooling) operation. For this experiment, as pooling operators we consider Adaptive Structure Aware Pooling (ASAP) (Ranjan et al., 2020), $k$-Maximal Independent Sets pooling ($k$-MIS) (Bacciu et al., 2023), MaxCut (Abate & Bianchi, 2025), Node Decimation Pooling (NDP) (Bianchi et al., 2020b), and Top-$k$. Since in this task we are considering larger graphs, we omit the dense operators presented in Section 4.1, which are characterized by a higher memory and time complexity. The datasets and the experimental setup are described in Appendices C.3 and D.2, respectively.

Table 2: Node classification. We report accuracy (Amazon-ratings, Roman-empire) and AUROC (Minesweeper, Tolokers, Questions). For each dataset: best performing pooler (**bold**); second best (underlined). Score: sum of points (2 for best, 1 for second best) across datasets.

| Pooler | Amazon-ratings | Minesweeper | Questions | Roman-empire | Tolokers | Score |
|--------|----------------|-------------|-----------|--------------|----------|-------|
| ASAP   | $\underline{43}_{\pm 1}$ | $\mathbf{79}_{\pm 0}$ | $\underline{66}_{\pm 3}$ | $34_{\pm 2}$ | $\mathbf{80}_{\pm 1}$ | $\underline{6}$ |
| $k$-MIS | $42_{\pm 2}$ | $57_{\pm 1}$ | $56_{\pm 7}$ | $28_{\pm 1}$ | $51_{\pm 3}$ | $0$ |
| MaxCut | $\underline{43}_{\pm 2}$ | $\underline{74}_{\pm 1}$ | $\mathbf{67}_{\pm 2}$ | $\mathbf{50}_{\pm 2}$ | $\underline{79}_{\pm 1}$ | $\mathbf{7}$ |
| NDP    | $\mathbf{44}_{\pm 0}$ | $71_{\pm 1}$ | $\underline{66}_{\pm 2}$ | $\underline{39}_{\pm 1}$ | $73_{\pm 4}$ | $4$ |
| Top-$k$ | $\underline{43}_{\pm 1}$ | $69_{\pm 1}$ | $\underline{66}_{\pm 4}$ | $37_{\pm 3}$ | $74_{\pm 4}$ | $2$ |

Table 2 indicates that performance is influenced by two key factors: an effective inductive bias and trainability. For instance, the inductive bias of NDP to sample the graph uniformly is particularly effective in heterophilic settings. In contrast, ASAP achieves even better results through its ability to learn a flexible, task-specific selection of nodes. The top performer, MaxCut, effectively combines the best of two worlds. The maxcut objective provides a bias towards uniform sampling solutions and, thanks to its trainability, also account for the specific requirements of the classification task.

## 4.3 GRAPH-LEVEL CLASSIFICATION AND REGRESSION

This experiment benchmarks several pooling operators on graph-level classification and regression. The goal is to assess their ability to learn coarsened graph representations that effectively summarize graph-level properties. In this experiment, we consider all kinds of operators (dense, sparse, trainable, and non-trainable): ASAP, Bayesian Nonparametric Pooling (BNPool) (Castellana & Bianchi, 2025), Edge-Contraction Pooling (ECPool) (Diehl, 2019; Landolfi, 2022), GraClus (Dhillon et al., 2007), Laplacian Pooling (LaPool) (Noutahi et al., 2019), MaxCut, MinCut, Self-Attention Graph pooling (SAG) (Lee et al., 2019), Top-$k$, and Path integral based pooling (PAN) (Ma et al., 2020). We refer to Appendices C.3 and D.3 for details about datasets and experimental setup.

The results in Table 3 demonstrate that **there is no "one-size-fits-all" pooling operator**. Instead, performance is highly dependent on the alignment between a pooler's inductive bias and the dataset's structural properties. This is particularly evident on the peptide datasets (pep-f and pep-s), where dense clustering operators like ACC, MinCut, and HOSC consistently deliver the best results. Their ability to detect node communities seems better suited for capturing the functional motifs present in molecular structures. On the other hand, MaxCut dramatically outperforms all competitors on Multipartite thanks to its inductive bias of retaining disconnected nodes, which aligns perfectly with the heterophilic structure of the dataset (Abate & Bianchi, 2025). These findings underscore that the choice of a pooling strategy is not arbitrary but should be an informed decision, guided by the characteristics of the graphs and the task requirements.

Table 3: Graph classification and regression. We report ROC-AUC for molhiv, MAE for pep-s, AP for pep-f and classification accuracy for the others. For each dataset: best performing pooler (**bold**); second best (underlined). Rank: sum of points (2 for best, 1 for second best) across datasets.

| Pooler | GCB-H | EXPWL1 | Multipartite | NCI1 | Reddit-b | molhiv | pep-f | pep-s | Score |
|--------|-------|--------|--------------|------|----------|--------|-------|-------|-------|
| ASAP | $67 \pm 4$ | $87 \pm 2$ | $24 \pm 16$ | $74 \pm 2$ | $87 \pm 3$ | $\underline{76} \pm 1$ | $72 \pm 0$ | $0.32 \pm 0.01$ | 1 |
| BNPool | $67 \pm 3$ | $71 \pm 2$ | $53 \pm 2$ | $76 \pm 1$ | $88 \pm 2$ | $\mathbf{77} \pm \mathbf{0}$ | $73 \pm 0$ | $0.29 \pm 0.00$ | 3 |
| ECPool | $\underline{72} \pm 1$ | $89 \pm 2$ | $51 \pm 2$ | $76 \pm 2$ | $\underline{90} \pm 3$ | $75 \pm 1$ | $72 \pm 0$ | $0.31 \pm 0.00$ | 2 |
| GraClus | $\underline{72} \pm 1$ | $91 \pm 2$ | $48 \pm 2$ | $76 \pm 2$ | $\underline{90} \pm 2$ | $74 \pm 0$ | $72 \pm 0$ | $0.31 \pm 0.00$ | 2 |
| LaPool | $70 \pm 2$ | $87 \pm 2$ | $20 \pm 16$ | $77 \pm 2$ | $\underline{90} \pm 2$ | $74 \pm 0$ | $71 \pm 0$ | $0.30 \pm 0.00$ | 1 |
| MaxCut | $71 \pm 3$ | $\mathbf{100} \pm \mathbf{0}$ | $\mathbf{79} \pm \mathbf{2}$ | $75 \pm 2$ | $86 \pm 2$ | $70 \pm 2$ | $68 \pm 1$ | $0.37 \pm 0.02$ | 4 |
| MinCut | $70 \pm 1$ | $71 \pm 2$ | $56 \pm 3$ | $75 \pm 2$ | $87 \pm 2$ | $75 \pm 1$ | $\mathbf{75} \pm \mathbf{0}$ | $\underline{0.29} \pm 0.00$ | 3 |
| PAN | $55 \pm 15$ | $72 \pm 5$ | $10 \pm 0$ | $71 \pm 2$ | $83 \pm 3$ | $71 \pm 1$ | $68 \pm 1$ | $0.31 \pm 0.00$ | 0 |
| SAG | $64 \pm 8$ | $74 \pm 6$ | $49 \pm 5$ | $72 \pm 2$ | $84 \pm 3$ | $72 \pm 1$ | $71 \pm 0$ | $0.32 \pm 0.01$ | 0 |
| Top-$k$ | $58 \pm 7$ | $52 \pm 7$ | $45 \pm 3$ | $72 \pm 2$ | $80 \pm 2$ | $\underline{76} \pm 1$ | $71 \pm 0$ | $0.31 \pm 0.00$ | 1 |
| $k$-MIS | $\mathbf{73} \pm \mathbf{1}$ | $\mathbf{100} \pm \mathbf{0}$ | $\underline{63} \pm 2$ | $77 \pm 2$ | $87 \pm 3$ | $\underline{76} \pm 2$ | $73 \pm 0$ | $0.30 \pm 0.00$ | 6 |
| ACC | $\underline{72} \pm 2$ | $93 \pm 1$ | $62 \pm 3$ | $\underline{78} \pm 2$ | $\mathbf{91} \pm \mathbf{2}$ | $75 \pm 1$ | $\mathbf{75} \pm \mathbf{0}$ | $\underline{0.29} \pm 0.00$ | $\underline{7}$ |
| NDP | $\underline{72} \pm 1$ | $98 \pm 1$ | $59 \pm 1$ | $77 \pm 2$ | $88 \pm 1$ | $75 \pm 1$ | $72 \pm 0$ | $0.30 \pm 0.00$ | 1 |
| HOSC | $71 \pm 1$ | $94 \pm 2$ | $\underline{63} \pm 2$ | $\underline{78} \pm 2$ | $\mathbf{91} \pm \mathbf{2}$ | $\underline{76} \pm 0$ | $73 \pm 0$ | $\mathbf{0.28} \pm \mathbf{0.00}$ | $\underline{7}$ |
| JBPool | $\mathbf{73} \pm \mathbf{1}$ | $\underline{99} \pm 1$ | $58 \pm 2$ | $\mathbf{79} \pm \mathbf{2}$ | $\mathbf{91} \pm \mathbf{2}$ | $73 \pm 1$ | $69 \pm 0$ | $0.29 \pm 0.00$ | **8** |
| DMoN | $71 \pm 1$ | $95 \pm 2$ | $62 \pm 3$ | $\underline{78} \pm 2$ | $\mathbf{91} \pm \mathbf{2}$ | $75 \pm 2$ | $\underline{74} \pm 0$ | $0.29 \pm 0.00$ | 5 |
| Diffpool | $65 \pm 2$ | $97 \pm 1$ | $56 \pm 3$ | $\mathbf{79} \pm \mathbf{2}$ | $\mathbf{91} \pm 1$ | $74 \pm 1$ | $66 \pm 1$ | $0.33 \pm 0.00$ | 4 |

## 4.4 Efficiency of pre-coarsening and caching

This experiment quantifies the computational speedup provided by the library's efficiency features for non-trainable pooling operators. We measure the average training time per epoch with and without caching for single-graph tasks, and with and without pre-coarsening for multi-graph tasks. For this experiment, we evaluate three deterministic pooling operators: $k$-MIS (with a non-trainable scoring function), NDP, and Non-negative Matrix Factorization pooling (NMF) (Bacciu & Di Sotto, 2019). Details about the experimental protocol are in Appendix D.4.

Table 4: Efficiency of caching (node classification) and pre-coarsening (graph classification). We report the average time to process a batch in the forward pass. For each operator, the second row reports times when using caching/pre-coarsening. N/C indicates that the algorithm failed to converge.

| Pooler | Amazon-r. | Roman-e. | Tolokers | GCB-H | MUTAG | Reddit-b |
|--------|-----------|----------|----------|-------|-------|----------|
| $k$-MIS | 0.008 | 0.008 | 0.009 | 0.006 | 0.005 | 0.007 |
| $k$-MIS (pre) | 0.004 | 0.004 | 0.004 | 0.001 | 0.001 | 0.001 |
| NDP | 13.16 | 7.33 | 72.02 | 3.42 | 0.96 | 7.07 |
| NDP (pre) | 0.004 | 0.004 | 0.009 | 0.001 | 0.001 | 0.001 |
| NMF | N/C | N/C | N/C | 3.45 | 1.14 | 210.0 |
| NMF (pre) | N/C | N/C | N/C | 0.001 | 0.001 | 0.002 |

The results in Table 4 show that the caching and pre-coarsening features lead to a dramatic, often orders-of-magnitude, reduction in training time. The speedup is particularly significant for operators with high computational complexity, such as NDP and NMF. This confirms that for deterministic poolers, `tgp`'s caching and pre-coarsening can transform computationally expensive methods into highly efficient operators, enabling their use in large-scale experiments and significantly accelerating operational pipelines.

## 4.5 Exploration of GNNs and Pooling Design

The modular design of `tgp` makes it straightforward to build non-standard pooling setups and deeper hierarchical GNNs. We illustrate this with two case studies: (i) swapping internal SRC(L) components

Table 5: Evaluation on advanced use cases. **Left:** Performance on NCI1 when swapping the default `SparseConnect` with `KronConnect`. **Right:** Performance on Reddit with increasing hierarchical depth (number of pooling layers).

| Pooler | SparseConnect | KronConnect |
|--------|---------------|-------------|
| Top-$k$ | $72 \pm 2$ | $73 \pm 2$ |
| $k$-MIS | $77 \pm 2$ | $70 \pm 2$ |

| Depth | MaxCut | MinCut |
|-------|--------|--------|
| 1 | $86 \pm 2$ | $87 \pm 2$ |
| 2 | $90 \pm 2$ | $91 \pm 2$ |
| 3 | $88 \pm 4$ | $91 \pm 2$ |

to create hybrid poolers; and (ii) stacking multiple pooling layers to vary hierarchical depth. As discussed in Section 3.2, `tgp` decouples the *Select*, *Reduce*, and *Connect* stages, so individual modules can be exchanged with a single line of code.

We assess the effect of changing only the connector by comparing standard Top-$k$ and $k$-MIS (both with `SparseConnect`) to variants where `KronConnect` is hot-swapped. Experiments are run on NCI1. This isolates the contribution of the *Connect* stage while leaving selection and reduction untouched. To test chaining multiple poolers within the same model, we train GNNs on Reddit that alternate MP layers with either MinCut or MaxCut, using architectures with 1, 2, or 3 pooling layers. This probes how performance changes as hierarchical depth increases.

Table 5 shows that (i) swapping only the connector can change accuracy on NCI1, underscoring the importance of isolating and studying specific pooling components; and (ii) on Reddit, performance depends on the number of pooling layers, indicating that hierarchical depth is a meaningful hyperparameter. In both scenarios, `tgp` enables controlled ablations and rapid reconfiguration of GNN blocks with minimal code changes. Details about the experimental setup can be found in Appendix D.3.

### 4.6 SUMMARY OF FINDINGS

Our benchmark leads to a clear conclusion: there is no universally superior pooling method. The optimal choice depends on how well the inductive bias of the pooler matches the data and the task at hand. In addition, the choice should be guided by trade-offs, such as the raw performance and computational costs. For example, clustering-based dense methods are best suited for homophilic graphs. In contrast, sparse and, particularly, non-trainable methods that can be precomputed are excellent choices for larger-scale applications or faster development cycles. In addition, non-trainable methods might have an advantage in smaller datasets, having the resulting GNN fewer parameters to fit and a stronger inductive bias. Finally, methods like JBPool and $k$-MIS achieve the worst performances in the node clustering and node classification tasks, but are among the best in the graph-level task. This reinforces the value of `tgp` as a toolkit for systematically exploring this diverse landscape to find the most suitable operator for a given problem.

### 5 CONCLUSION AND FUTURE DIRECTIONS

We introduced `tgp`, a comprehensive and extensible library that unifies the fragmented landscape of graph pooling. By implementing a wide variety of operators under a consistent, modular API inspired by the SRC(L) framework, `tgp` makes different pooling strategies fully interchangeable. This design significantly simplifies the development and comparative analysis of hierarchical GNNs implemented with PyG. From a practical perspective, `tgp` allows for fast prototyping. In addition, the caching and pre-coarsening features transform computationally expensive deterministic methods into handy, high-speed alternatives, enabling their use in large-scale applications. Scientifically, `tgp` lowers the barrier to entry for both research and development of hierarchical GNNs and enables fair, reproducible comparisons.

In the near future, we plan to include additional existing pooling operators and keep `tgp` up to date by incorporating new methods as they emerge. From a systems perspective, future work will focus on performance optimizations; this includes investigating more efficient implementations of core components, such as the Select operation for large graphs, and exploring approximate and distributed algorithms to improve scalability. We invite both researchers and practitioners to utilize `tgp`, contribute to it, for accelerating the research and development of hierarchical GNNs.

ETHICS STATEMENT

The work presented in this paper introduces a general-purpose software library and performs benchmarks on standard, publicly available datasets. The authors have read and adhere to the ICLR Code of Ethics and do not foresee any direct ethical concerns or potential for misuse.

REPRODUCIBILITY STATEMENT

The code of Torch Geometric Pool and the code to reproduce all the experiments presented in this paper are open-source and included in the supplementary material. Everything will be made publicly available upon acceptance. Details regarding the model architectures, hyperparameters, and training protocols for each benchmark are provided in the Appendix.

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

## A    IMPLEMENTED POOLING LAYERS

Torch Geometric Pool provides a wide array of pooling operators from the literature, covering all branches of the hierarchical pooling taxonomy. The available poolers are Adaptive Structure Aware Pooling (ASAP), Asymmetric Cheeger Cut pooling (ACC), Bayesian Nonparametric Pooling (BNPool), Deep Modularity Networks (DMoN), , Edge-Contraction Pooling (ECPool), , High-Order Spectral Clustering (HOSC), Just-Balance Pooling (JBPool), $k$-Maximal Independent Sets pooling ($k$-MIS), Laplacian Pooling (LaPool), (), Node Decimation Pooling (NDP), Non-negative Matrix Factorization pooling (NMF), Path integral based pooling (PAN), Self-Attention Graph pooling (SAG), and  (). Table 6 summarizes the key characteristics of the implemented operators, highlighting whether a method is sparse or dense, trainable or not, and if it provides auxiliary losses.

Table 6: Key characteristics of the pooling operators currently implemented in our library. The checkmarks indicate if an operator produces a sparse assignment matrix, has trainable parameters, and/or uses an auxiliary loss term for optimization.

| Operator | Sparse | Trainable | Auxiliary Loss |
|---|:---:|:---:|:---:|
| ASAP (Ranjan et al., 2020) | ✓ | ✓ | |
| ACC (Hansen & Bianchi, 2023) | | ✓ | ✓ |
| BNPool (Castellana & Bianchi, 2025) | | ✓ | ✓ |
| DMoN (Tsitsulin et al., 2023) | | ✓ | ✓ |
| Diffpool (Ying et al., 2018) | | ✓ | ✓ |
| ECPool (Diehl, 2019; Landolfi, 2022) | ✓ | ✓ | |
| GraClus (Dhillon et al., 2007) | ✓ | | |
| HOSC (Duval & Malliaros, 2022) | | ✓ | ✓ |
| JBPool (Bianchi, 2022) | | ✓ | ✓ |
| $k$-MIS (Bacciu et al., 2023) | ✓ | | |
| LaPool (Noutahi et al., 2019) | ✓ | | |
| MaxCut (Abate & Bianchi, 2025) | ✓ | ✓ | ✓ |
| MinCut (Bianchi et al., 2020a) | | ✓ | ✓ |
| NDP (Bianchi et al., 2020b) | ✓ | | |
| NMF (Bacciu & Di Sotto, 2019) | | | |
| PAN (Ma et al., 2020) | ✓ | ✓ | |
| SAG (Lee et al., 2019) | ✓ | ✓ | |
| Top-$k$ (Gao & Ji, 2019; Knyazev et al., 2019) | ✓ | ✓ | |

## B    STANDARDIZED DATA STRUCTURES

To enforce modularity and provide a consistent API, `tgp` relies on two key data structures that act as standardized interfaces.

**The `SelectOutput` class**   is the standardized return type for all `Select` modules. It serves as a container for the assignment matrix $S$ and related information, such as the indices of selected nodes. This decouples the logic of the Select operator from those of the Reduce and Connect, which can operate on the `SelectOutput` object without needing to know how Select was implemented.

Beyond being a simple container, `SelectOutput` is equipped with powerful utility methods. A notable example is `assign_all_nodes()`, which can algorithmically extend a sparse node selection (like that from Top-$k$) into a full partition, assigning every node in the graph to its nearest supernode. This provides a simple way to bridge the gap between the output of score-based methods and dense poolers based on soft-clustering.

**The `PoolingOutput` class** is the standardized return type for the `forward()` call of every pooling operator in `tgp`. It consistently bundles all the necessary information for downstream layers, including:

- The pooled node features $X'$.

- The coarsened graph structure $A'$ (`edge_index` and `edge_weights`).

- The updated `batch` vector for the pooled nodes.

- The original `SelectOutput` object, for use in lifting operations or analysis.

- A dictionary of `loss` terms, if the pooler computes an auxiliary loss.

The `PoolingOutput` object also provides convenient helper properties and methods, such as `has_loss` and `get_loss_value()`, which simplify the process of incorporating auxiliary losses into the main training loop.

## C    DATASET DETAILS

### C.1    NODE CLUSTERING DATASETS

For the unsupervised clustering task, we use four well-known citation network benchmarks (Yang et al., 2016; Fu et al., 2020) and a synthetic dataset designed to provide a controlled evaluation environment. The citation network datasets are loaded using the API provided by PyG. The synthetic Community dataset is generated with PyGSP (Defferrard et al., 2017) and consists of a graph sampled from a stochastic block model with 5 communities. The statistics for all datasets are reported in Table 7.

Table 7: Details of the vertex clustering datasets.

| Dataset | #Vertices | #Edges | #Vertex attr. | #Classes |
|---|---|---|---|---|
| Community | 400 | 5,904 | 2 | 5 |
| Cora | 2,708 | 10,556 | 1,433 | 7 |
| Citeseer | 3,327 | 9,104 | 3,703 | 6 |
| Pubmed | 19,717 | 88,648 | 500 | 3 |
| DBLP | 17,716 | 105,734 | 1,639 | 4 |

### C.2    NODE CLASSIFICATION DATASETS

The node classification experiments are conducted on the five large-scale heterophilic graphs introduced by (Platonov et al., 2023). These datasets are loaded using the API provided by PyG and come with 10 predefined public splits for training, validation, and testing. The statistics of the five datasets are reported in Table 8. The column $h(\mathcal{G})$ reports the class-insensitive edge homophily ratio (Lim et al., 2021), where lower values indicate a higher degree of heterophily.

Table 8: Statistics of the heterophilic node classification datasets.

| Dataset | # Nodes | # Edges | # Classes | $h(\mathcal{G})$ |
|---|---|---|---|---|
| Roman-Empire | 22,662 | 32,927 | 18 | 0.021 |
| Amazon-Ratings | 24,492 | 93,050 | 5 | 0.127 |
| Minesweeper | 10,000 | 39,402 | 2 | 0.009 |
| Tolokers | 11,758 | 519,000 | 2 | 0.180 |
| Questions | 48,921 | 153,540 | 2 | 0.079 |

### C.3 GRAPH-LEVEL TASKS DATASETS

For the benchmark on graph-level tasks, we consider graph classification and graph regression datasets from diverse domains and with different graph characteristics. In particular, we included in our evaluation bioinformatics datasets from the TUDatasets collection (Morris et al., 2020) (NCI1), social networks (Reddit-binary), and synthetic benchmarks designed to test specific model capabilities (expwl1 (Bianchi & Lachi, 2023), Multipartite (Abate & Bianchi, 2025), Graph Classification Benchmark Hard (GCB-H) (Bianchi et al., 2022)). Additionally, we use large-scale molecular datasets including peptides-func and peptides-struct from the Long-Range Graph Benchmark (LRGB) collection (Dwivedi et al., 2022), and ogbg-molhiv from the protein-protein association networks (ogbg-ppa) (Hu et al., 2020). The statistics for each dataset are provided in Table 9.

Table 9: Details of selected graph classification datasets.

| Dataset | # Graphs | # Classes | Avg. # Nodes | Avg. # Edges |
|---|---|---|---|---|
| NCI1 | 4,110 | 2 | 29.87 | 64.60 |
| Reddit-binary | 2,000 | 2 | 429.63 | 497.75 |
| EXPWL1 | 3,000 | 2 | 76.96 | 186.46 |
| Multipartite | 5,000 | 10 | 99.79 | 4,477.43 |
| ogbg-molhiv | 41,127 | 2 | 25.5 | 27.5 |
| peptides-func | 15,535 | 10 | 150.94 | 307.30 |
| peptides-struct | 15,535 | 11 | 150.94 | 307.30 |
| GCB-H | 1,800 | 3 | 148.32 | 572.32 |

## D EXPERIMENTAL SETUPS

### D.1 UNSUPERVISED NODE CLUSTERING

The GNN architecture consists of a simple encoder followed by a trainable and dense pooling layer. The encoder is composed of two MP operators implemented as `ARMAConv` layers (Bianchi et al., 2021) with ELU activation functions, which map the input node features to a 32-dimensional embedding space. This learned embedding is then fed directly into the pooling operator (see Figure 3). The

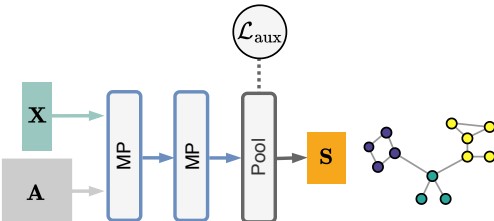

Figure 3: Node clustering architecture.

entire GNN is trained exclusively on the auxiliary losses provided by `PoolingOutput`. The model is trained for up to 2000 epochs using the Adam optimizer (Kingma & Ba, 2014) with a learning rate of $5 \cdot 10^{-4}$ and the ReduceLROnPlateau scheduler[2], with early stopping (patience of 500 epochs). After training, we extract hard cluster assignments via an $\mathrm{argmax}$ operation on the matrix $S$ and evaluate quality using the Normalized Mutual Information (NMI).

### D.2 NODE CLASSIFICATION

We adopt a hierarchical autoencoder architecture (similar to the Graph U-Net architecture (Gao & Ji, 2019)) with the structure: MP-Pool-MP-Unpool-MP-Readout. As the MP layer, we used Graph

---

[2]https://docs.pytorch.org/docs/stable/generated/torch.optim.lr_scheduler.ReduceLROnPlateau.html

Isomorphism Network (GIN) (Xu et al., 2019) with 32 hidden units and ReLU activation. The readout is implemented as a Multilayer Perceptron (MLP) with one hidden layer and a dropout rate of $0.1$. A schematic depiction of the architecture is reported in Figure 4. The model is trained for up to 20,000

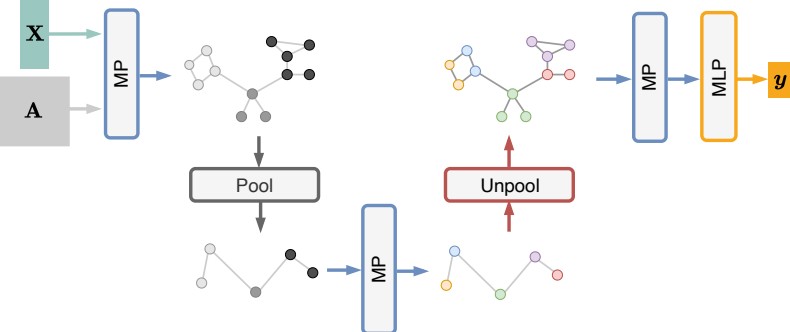

Figure 4: Node classification architecture.

epochs by jointly minimizing the cross-entropy loss and any auxiliary loss, using the Adam optimizer with a learning rate of $5 \cdot 10^{-4}$, a scheduler, and early stopping (patience of 2,000 epochs). For each dataset, we report the performance in terms of Accuracy or ROC AUC over 10 public data folds, following the same setting proposed in the original paper (Platonov et al., 2023).

### D.3 GRAPH-LEVEL TASKS

The model architecture is a hierarchical GNN with a GIN layer (32 hidden channels, `ELU` activation), both before and after the pooling operation. The model leverages the pooler's is_dense attribute to conditionally switch between a standard and dense version of GIN (`DenseGINConv` layer[3]) in the post-pooling block. The readout consists of a global sum pooling followed by a 3-layer MLP with a dropout of $0.5$. In the deeper GNN architecture with multiple pooling layers presented in Section 4.5, the same [`Pool - MP`] block has been replicated multiple times.

A schematic depiction of the architecture is reported in Figure 5. For graph classification, we optimize

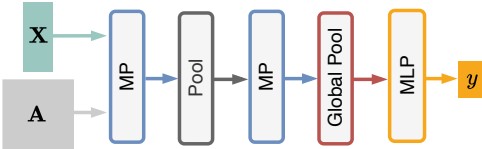

Figure 5: Architecture for graph-level tasks (graph classification and regression).

either the cross-entropy or the binary cross-entropy; for graph regression, we minimize the mean squared error. If a pooler provides one or more auxiliary losses, these are jointly optimized with the task loss by summing all losses together. The models are trained for up to 1000 epochs using the Adam optimizer with a learning rate of $1 \cdot 10^{-4}$, a scheduler, and early stopping (patience of 300 epochs). For datasets with public splits, we perform 10 runs; otherwise, we use 10-fold cross-validation. For the results, we report the test classification accuracy, except for ogbg-molhiv (ROC AUC), pep-function (average precision), and pep-struct (mean absolute error).

### D.4 EFFICIENCY ANALYSIS

The efficiency analysis was designed to measure the reduction in training time provided by `tgp`'s caching and pre-coarsening features for non-trainable, deterministic pooling operators. The experiments were conducted on an NVIDIA RTX 6000 Ada Generation by measuring the average epoch time (in seconds) over 100 epochs.

---

[3]https://pytorch-geometric.readthedocs.io/en/2.5.0/generated/torch_geometric.nn.dense.DenseGINConv.html

Figure 6: Architecture for graph-level tasks with pre-coarsening.

We evaluated two distinct scenarios:

- **Caching for Node Classification:** On single-graph datasets, the caching mechanism computes the coarsened graph structure once and reuses it for all subsequent epochs. The architecture is the same as in Appendix D.2. Caching is tested on the node classification datasets Amazon-ratings, Roman-empire and Tolokers.

- **Pre-coarsening for Graph Classification:** For datasets containing multiple graphs, the pre-coarsening is applied once before training as a pre-transform. For this experiment we used a hierarchical GNN with 3 convolutional layers alternating with 2 pooling layers, employing GIN convolutions with 32 hidden channels and an ELU activation function, followed by a dropout of 0.5. The model was trained for 100 epochs with a batch size of 32 using the Adam optimizer with an initial learning rate of $10^{-4}$. A schematic depiction of the architecture is reported in Figure 6. Pre-coarsening is evaluated on the graph-level datasets GCB-H, MUTAG and Reddit-binary.

# E ADDITIONAL RESULTS

Table 10: Clustering accuracy by different poolers on node clustering. For each dataset: best performing pooler (**bold**); second best (underlined).

| Pooler | CiteSeer | Community | Cora | DBLP | PubMed |
|---|---|---|---|---|---|
| ACC | **0.5162** $\pm$ **0.0502** | 0.9442 $\pm$ 0.0536 | **0.5324** $\pm$ **0.0470** | 0.5067 $\pm$ 0.0499 | 0.5729 $\pm$ 0.0445 |
| Diffpool | 0.3319 $\pm$ 0.0335 | 0.7660 $\pm$ 0.0116 | 0.3996 $\pm$ 0.0348 | 0.4554 $\pm$ 0.0142 | 0.4332 $\pm$ 0.0435 |
| DMoN | 0.4455 $\pm$ 0.0108 | **0.9903** $\pm$ **0.0008** | 0.4683 $\pm$ 0.0304 | 0.5234 $\pm$ 0.0162 | **0.5738** $\pm$ **0.0709** |
| HOSC | 0.3858 $\pm$ 0.0337 | 0.7587 $\pm$ 0.1242 | 0.3817 $\pm$ 0.0474 | **0.6231** $\pm$ **0.0174** | 0.5249 $\pm$ 0.0487 |
| JBPool | 0.3684 $\pm$ 0.0442 | 0.8962 $\pm$ 0.1347 | 0.3520 $\pm$ 0.0373 | 0.5048 $\pm$ 0.0670 | 0.5264 $\pm$ 0.0479 |
| MinCut | 0.4120 $\pm$ 0.0451 | 0.9517 $\pm$ 0.0021 | 0.4894 $\pm$ 0.0451 | 0.5201 $\pm$ 0.0519 | 0.5595 $\pm$ 0.0543 |

Table 11: F1 score (macro-averaged) by different poolers on node clustering. For each dataset: best performing pooler (**bold**); second best (underlined).

| Pooler | CiteSeer | Community | Cora | DBLP | PubMed |
|---|---|---|---|---|---|
| ACC | **0.4578** $\pm$ **0.0510** | 0.9376 $\pm$ 0.0762 | 0.4477 $\pm$ 0.0470 | 0.4612 $\pm$ 0.0546 | 0.5661 $\pm$ 0.0591 |
| Diffpool | 0.2729 $\pm$ 0.0296 | 0.7679 $\pm$ 0.0113 | 0.3583 $\pm$ 0.0478 | 0.3304 $\pm$ 0.0148 | 0.3814 $\pm$ 0.0592 |
| DMoN | 0.4181 $\pm$ 0.0097 | **0.9902** $\pm$ **0.0008** | 0.4348 $\pm$ 0.0296 | 0.4758 $\pm$ 0.0182 | **0.5703** $\pm$ **0.0739** |
| HOSC | 0.3086 $\pm$ 0.0336 | 0.7071 $\pm$ 0.1503 | 0.2780 $\pm$ 0.0550 | **0.5008** $\pm$ **0.0150** | 0.5164 $\pm$ 0.0541 |
| JBPool | 0.3211 $\pm$ 0.0482 | 0.8771 $\pm$ 0.1584 | 0.2750 $\pm$ 0.0496 | 0.4478 $\pm$ 0.0640 | 0.5311 $\pm$ 0.0501 |
| MinCut | 0.3949 $\pm$ 0.0402 | 0.9521 $\pm$ 0.0021 | **0.4710** $\pm$ **0.0458** | 0.4790 $\pm$ 0.0484 | 0.5577 $\pm$ 0.0608 |

We consider two additional metrics for the clustering task. Because cluster labels are permutation-invariant, we align predicted labels $\hat{y}_i$ to ground-truth $y_i$ by finding the permutation $\pi$ that maximizes agreement via the Hungarian (Munkres) algorithm. The cluster accuracy is defined as:

$$\text{ClustAcc} = \frac{1}{n} \sum_{i=1}^{n} \mathbf{1}\{y_i = \pi(\hat{y}_i)\}.$$

Given the same optimal permutation, precision and recall are computed per class $k$ as $\mathrm{Prec}_k = \frac{\mathrm{TP}_k}{\mathrm{TP}_k + \mathrm{FP}_k}$ and $\mathrm{Rec}_k = \frac{\mathrm{TP}_k}{\mathrm{TP}_k + \mathrm{FN}_k}$, and the per-class F1-score is

$$F1_k = \frac{2\,\mathrm{Prec}_k\,\mathrm{Rec}_k}{\mathrm{Prec}_k + \mathrm{Rec}_k}.$$

Tables 10 and 11 report respectively the cluster accuracy and the *macro-F1*, defined as the unweighted mean $\frac{1}{K}\sum_{k=1}^{K} F1_k$.

