# OpenReview forum: "Torch Geometric Pool: the Pytorch library for pooling in Graph Neural Networks"
_ICLR.cc/2026/Conference — Submitted to ICLR 2026_

### Official Review · Reviewer_osFv · 2025-10-20

**Soundness:** 3
**Presentation:** 3
**Contribution:** 1
**Rating:** 2
**Confidence:** 5

**Summary:**

This paper presents Torch Geometric Pool (tgp), a modular library for hierarchical pooling in Graph Neural Networks, built on top of PyTorch Geometric. The library unifies diverse pooling operators under a consistent Select-Reduce-Connect-Lift framework, emphasizing usability, extensibility, and computational efficiency through features like precomputed pooling. Comprehensive experiments across various downstream tasks demonstrate that the optimal pooling method is task-dependent, highlighting the importance of a flexible library for rapid prototyping and evaluation.

**Strengths:**

- The writing is good and easy to read.
- A good contribution to the GNN community.

**Weaknesses:**

- The contribution is primarily engineering-oriented and lacks sufficient methodological innovation.
- The current implementation only supports homophilic graphs, which limits its applicability to broader graph types.
- The experimental design is not fully consistent with the stated goal of evaluating hierarchical pooling approaches.

**Questions:**

- The package is titled “Torch Geometric Pool: the PyTorch library for pooling in Graph Neural Networks”, yet it focuses solely on hierarchical pooling and omits global pooling methods. The title may therefore overstate the scope of the library.
- The in-memory caching mechanism is designed for single-graph tasks but may restrict scalability for large graphs. How does the framework handle such scalability concerns?
- The experimental evaluation is not comprehensive, several included pooling operators are not compared across all relevant tasks.
- In the clustering experiments, dense pooling methods produce fully connected weighted graphs. How is the threshold determined for forming clusters?
- Metrics such as ACC, ARI and F1-score should be included for clustering evaluation.
- This paper targets hierarchical pooling, but the downstream GNN architectures for clustering, node classification, and graph classification use only a single pooling layer, which does not sufficiently demonstrate the hierarchical nature of these compared pooling methods.
- Given the engineering focus and practical utility, this work may be more suitable for a workshop submission, similar to prior PyG 1.0 and 2.0 releases. Alternatively, the authors could consider contributing it as a pull request to the PyG project, which would enhance accessibility and impact within the community.
- Other graph hierarchical pooling works:
	* HGP-SL. Zhang Z, Bu J, Ester M, et al. Hierarchical multi-view graph pooling with structure learning[J]. IEEE Transactions on Knowledge and Data Engineering, 2021, 35(1): 545-559.
	* GSAPool. Zhang L, Wang X, Li H, et al. Structure-feature based graph self-adaptive pooling[C]//Proceedings of the web conference 2020. 2020: 3098-3104.
	* SEP. Wu J, Chen X, Xu K, et al. Structural entropy guided graph hierarchical pooling[C]//International conference on machine learning. PMLR, 2022: 24017-24030.
	* GPN. Song Y, Huang S, Wang X, et al. Graph Parsing Networks[C]//The Twelfth International Conference on Learning Representations.
	* CGIPool. Pang Y, Zhao Y, Li D. Graph pooling via coarsened graph infomax[C]//Proceedings of the 44th international ACM SIGIR conference on research and development in information retrieval. 2021: 2177-2181.

---

> ### Author Response · Authors · 2025-11-19
> **Rebuttal (1/2)**
>
> Thanks for the review. Before answering each question in detail, we would like to comment that many of the reviewers’ criticism focuses on design choices of the experimental evaluation: metrics considered, number of layers in the architecture, combinations of poolers/datasets, and so on. We believe these are not necessarily aligned with the goals we set for our experimental evaluation:
>
> - Showcasing the functionalities and flexibility of `tgp` in the principal downstream tasks where pooling is used.
> - Showing that the best pooling method is not the same for each task and dataset at hand, meaning that one has to try several alternatives each time. This makes `tgp` an essential tool to enable this exploration.
>
> ## W1: Engineering-oriented contribution
>
> Please, refer to the general Official Comment
>
> ## W2: Homophilic graphs
>
> This is not correct. Our library's applicability is not limited to homophilic graphs.
>
> In fact, our entire node classification experiment (**Section 4.2**) was conducted on five large-scale heterophilic graphs. As detailed in **Appendix C.2**, these datasets were chosen specifically for this property, and our analysis even discusses how the inductive biases of certain poolers are particularly well-suited for these heterophilic settings.
>
> ## W3: Experimental design
>
> Please refer to the first part of the answer to your review and to the next points.
>
> ## Q1: focus on hierarchical pooling and omission of global pooling
>
> We appreciate the chance to clarify our scope and terminology.
>
> In the GNN community, the term "graph pooling" is very often used to refer to *hierarchical* pooling. This is because it involves a much more complex graph-to-graph operation, as opposed to global pooling, which simply aggregates the node features into a single vector. In the revised manuscript, we made sure to be clear from the beginning that this is the focus of `tgp`.
>
> Second, `tgp` *does* provide standardized global readout methods. A key feature of our unified API is the `global_pool()` method, which is available for every operator to abstract away the differences between sparse and dense data representations and produce a final graph embedding. This is described in **Section 3.1** (unified readout).
>
> Finally, we note that more sophisticated global operations are typically referred to as "Aggregators" in PyTorch Geometric, and a wide variety of these are already standardized and available in the core PyG library (https://pytorch-geometric.readthedocs.io/en/latest/modules/nn.html#aggregation-operators). Clearly, reimplementing them in `tgp` would be unnecessary and redundant. We clarified this point in **Section 3.1** of the revised manuscript.
>
> ## Q2: in-memory caching for large graphs
>
> The in-memory caching is designed as an optional convenience for static graphs where memory is not a constraint. For larger graphs, `tgp` provides a robust scalability solution via the **pre-coarsening mechanism** . It is crucial to distinguish between the peak memory required to *compute* the pooling operation (which can be high due to intermediate calculations) versus the memory required to *store* the result. Since the pooled graph is coarsened, it is strictly smaller than the original; therefore, if a GPU can hold the original graph, it can certainly hold the pooled one. The pre-coarsening mechanism leverages this by allowing the user to perform the memory-intensive transformations offline, i.e., on the RAM or on virtual memory (swap), and save the results to disk. Consequently, the training process occurring on a GPU with limited VRAM only needs to load these pre-computed, smaller structures, effectively offloading the heavy computational requirements of the pooling algorithm from the resources available during training.
>
> We have clarified this distinction and workflow in Section 3.3 of the revised manuscript.

---

> > ### Author Response · Authors · 2025-11-19
> > **Rebuttal (2/2)**
> >
> > ## Q3: pooling operators not compared across all tasks
> >
> > Our original experimental design was focused on demonstrating the core conclusion that there is **no 'one-size-fits-all' pooling operator**, a point we believe our initial findings sufficiently established.
> >
> > Crucially, we note that not all operators are conceptually or functionally applicable to each downstream task. For instance, in the unsupervised clustering experiment (**Section 4.1**), it is only possible to evaluate poolers that provide an auxiliary loss, which is the sole training signal; methods without such a loss cannot be used here. Similarly, node classification is evaluated on large graphs where dense pooling operators do not scale.
> >
> > To address the reviewer's concern regarding comprehensiveness, we leveraged the power of `tgp`'s architecture to include **all** implemented pooling operators in the graph classification and regression tasks (**Section 4.3**). This expanded, comprehensive evaluation serves two purposes:
> >
> > 1. It further substantiates our core finding that the optimal pooler choice is critically task- and data-dependent.
> > 2. It serves as a strong, practical demonstration of `tgp`'s unified API and modularity, proving that such a vast, complex benchmark—which would be highly cumbersome without our library—can now be performed easily and seamlessly.
> >
> > ## Q4: threshold to form clusters
> >
> > We believe there is a misunderstanding, as the clusters are not derived from the resulting fully connected weighted graph ($\mathbf{A}'$). Instead, the cluster assignments are determined directly by the $N \times K$ assignment matrix $\mathbf{S}$, which is the output of the `Select` operation. This matrix represents the "soft" membership of the $N$ original nodes to the $K$ supernodes, and these supernodes *are* the clusters.
> > To obtain the final hard cluster assignments for evaluation, we simply take an `argmax` on each row of $\mathbf{S}$, assigning each node to the cluster for which it has the highest membership score. So, no thresholding is required.
> >
> > ## Q5: additional clustering metrics
> >
> > We included F1 and ACC in Appendix E. ARI is basically the same as NMI, which was already present in the paper.
> >
> > ## Q6: single pooling layer
> >
> > Our experimental architectures, as shown in Figures 4 and 5, are indeed hierarchical as they explicitly learn multi-scale representations by coarsening the graph.
> >
> > We already demonstrated the library's ability to handle deeper hierarchical structures. Indeed, the architecture used in our efficiency analysis (**Section 4.4** and **Appendix D.4**) already incorporated two pooling layers.
> >
> > Furthermore, we have now added a new experiment in **Section 4.5** explicitly showcasing the ease of implementing a deep hierarchical network with various numbers of pooling layers, thereby validating `tgp`'s capacity to build arbitrarily deep hierarchical models effortlessly.
> >
> > ## Q7: engineering focus of the paper
> >
> > Please refer to the general Official Comment about the ICLR CfP
> >
> > ## Q8: Other graph hierarchical pooling works
> >
> > We thank the reviewer for providing this list of relevant methods. This highlights the active research in this area and reinforces the need for a standardized framework. `tgp` is designed to be extensible, and as we state in our conclusion, we plan to keep the library up-to-date by incorporating new methods as they emerge. We will also welcome and support community contributions to help accelerate this process.

---

> > > ### Comment · Reviewer_osFv · 2025-11-26
> > > **Thanks for the rebuttal**
> > >
> > > Q1. First, for what 'graph pooling' refers to, please refer to the literature reviews in [1–2].
> > >
> > > Second, although tgp supports global pooling, the abstract, introduction, and experimental design only focus on hierarchical pooling.
> > >
> > > Third, the term *Aggregators* in PyTorch Geometric refers to message-passing aggregation, not pooling. In contrast, *Pooling Layers* in PyG include both global and hierarchical pooling methods.
> > >
> > > [1] *Graph pooling in graph neural networks: methods and their applications in omics studies*
> > >
> > > [2] *Graph pooling for graph-level representation learning: a survey*
> > >
> > > Q3. If the experiments are not sufficiently comprehensive, it is hard to conclude that “there is no one-size-fits-all pooling operator.”
> > >
> > > Q6. Figures 4 and 5 do not represent hierarchical pooling. For architectures with hierarchical pooling for graph classification and node classification, please refer to MinCutPool [3] and gPool [4]. The experimental validation is not convincing.
> > >
> > > [3] *Spectral Clustering with Graph Neural Networks for Graph Pooling*
> > >
> > > [4] *Graph U-Nets*
> > >
> > > Q7. Besides above concerns, in my view, this work does not demonstrate contributions beyond those in PyG (1.0 or 2.0). Therefore, it remains unclear how this work fits the standards and expectations of ICLR.

---

> > > > ### Author Response · Authors · 2025-12-01
> > > >
> > > > We believe that your comments are either incorrect or very subjective. In our opinion, the goal of a scientific discussion should be to provide constructive feedback rather than looking for arguments to reject the paper.
> > > >
> > > > ## Subjective criticism
> > > >
> > > > **Experimental evaluation not convincing/not sufficiently comprehensive**
> > > >
> > > > We addressed your concerns in the rebuttal, adding additional experiments and explaining why some combinations of pooler/dataset were unfeasible. Yet, you hadn’t acknowledged that. In the current form, the paper reports experiments on 17 pooling operators on 18 different datasets. We have 30 experiments for node clustering, 25 for node classification, 136 (!) for graph-level tasks, 36 for the efficiency study in 4.4, and 10 in the new section 4.5 made to satisfy **your** request. Each experiment was repeated 10 times, giving a total of 2370 runs. This is a way larger experimental evaluation than most papers on GNNs and saying that it is “not sufficiently comprehensive”, especially given that this is a paper about a library, is completely unreasonable.
> > > >
> > > > We also note that there are no objective reasons why this experimental evaluation should not serve the stated purposes. In fact, **you are not providing any**.
> > > >
> > > >
> > > > **Contributions limited to PyG**
> > > >
> > > > Except for a few exceptions, the GNN research is nowadays done through PyG. There are no objective reasons (and, indeed, **you do not provide any**) why targeting the most popular and broadly used framework should not fit “the standards and expectations of ICLR”.
> > > >
> > > >
> > > > ## Incorrect statements
> > > >
> > > > **PyG aggregators**
> > > >
> > > > Aggregators in PyG are also used to perform global pooling. Notably, some of them were specifically proposed and designed for global pooling. See for example
> > > >
> > > > https://pytorch-geometric.readthedocs.io/en/latest/generated/torch_geometric.nn.aggr.SortAggregation.html#torch_geometric.nn.aggr.SortAggregation
> > > >
> > > > and
> > > >
> > > > https://pytorch-geometric.readthedocs.io/en/latest/generated/torch_geometric.nn.aggr.GraphMultisetTransformer.html#torch_geometric.nn.aggr.GraphMultisetTransformer
> > > >
> > > > Note that in their documentation they are referred to as global pooling operators.
> > > >
> > > > **Architectures with hierarchical pooling**
> > > >
> > > > The architecture for graph-level tasks is **identical** to the one used in MinCut. For node-level tasks in 4.2 the architecture is the same as the g-unet, except for the removal of the skip connections, which might allow bypassing pooling. Notably, such skip connections can be added to the architecture by setting a flag (see uploaded code).

---

### Official Review · Reviewer_4FU5 · 2025-10-26

**Soundness:** 3
**Presentation:** 3
**Contribution:** 2
**Rating:** 4
**Confidence:** 4

**Summary:**

The paper presents a PyTorch Geometric–based library that consolidates a broad set of graph pooling operators behind a consistent interface, organized by a select–reduce–connect–lift abstraction. It standardizes batching, readout, and auxiliary-loss handling; adds engineering optimizations such as caching and dataset-level pre-coarsening to speed up deterministic operators; and offers a systematic, within-library benchmark across tasks to illuminate trade-offs among sparse vs. dense and trainable vs. non-trainable pooling methods.

**Strengths:**

1. **Well-scoped unification that improves day-to-day usability.** Despite focusing on a single GNN subcomponent, the library makes swapping and configuring pooling operators straightforward via a single, coherent API; the standardized handling of batching, readout, and auxiliary losses reduces integration overhead and enables more reliable ablations without chasing disparate third-party repositories.

2. **A clear, modular taxonomy that clarifies the design space.** By structuring pooling into select–reduce–connect–lift stages, the work offers both a conceptual lens and code-level modules that help practitioners reason about alternatives and assemble hybrids with minimal boilerplate, which in turn encourages systematic exploration rather than ad-hoc, one-off implementations.

3. **Practical efficiency and reproducibility gains within a unified stack.** The inclusion of caching and pre-coarsening accelerations for deterministic operators, together with standardized datasets and scripts, supports faster iteration and more consistent, apples-to-apples comparisons inside the same framework, aiding reproducibility and lowering the barrier to controlled empirical studies.

**Weaknesses:**

1. **The scope and community impact appear limited for a venue like ICLR.** While the library offers a tidy unification of graph pooling under a common API, it targets a single GNN subcomponent rather than a broadly enabling platform; in contrast, end-to-end frameworks (e.g., general-purpose GNN libraries) typically shift community practice more substantially. Consequently, the manuscript’s contribution feels incremental at the level of research impact, even if it may be practically helpful for users who frequently switch among pooling operators.

2. **The work reads primarily as curation and repackaging rather than methodological or systems innovation.** Most pooling operators already have public implementations scattered across original repositories or mainstream GNN stacks, and practitioners can usually incorporate them with moderate effort; therefore, the added value here seems to lie in consolidation, consistent interfaces, and engineering polish rather than in new algorithms, theory, or systems primitives. Although such consolidation is useful, the paper should better articulate what is fundamentally new beyond harmonization—e.g., what capabilities would be difficult or cumbersome to realize without this library.

3. **Head-to-head comparisons against existing community implementations are missing where they matter most.** The manuscript would benefit from systematic evaluations of API ergonomics (e.g., installation friction, lines of code to swap methods, configuration clarity) and runtime characteristics (e.g., throughput and memory) against widely used baselines such as native modules in established libraries and the original authors’ code; importantly, while the paper appears to benchmark multiple pooling operators within its own framework, this is not the same as demonstrating external parity or superiority, and a fair, apples-to-apples comparison outside the proposed stack is necessary to substantiate the claimed practical value.

**Questions:**

see weaknesses.

---

> ### Author Response · Authors · 2025-11-14
> **Rebuttal**
>
> Thank you for the review.
>
> ### Weakness 1
>
> We respectfully disagree with the assessment that the scope and community impact are limited. While `tgp` focuses on the pooling subcomponent, we argue this is a critical, high-impact area precisely because it is so fragmented and underdeveloped in existing "broadly enabling" GNN frameworks. The reviewer suggests the tool is helpful only for the niche user who "frequently switches" poolers. While this is certainly important for pooling experts who need to design new custom methods or perform systematic, fair comparisons, we also argue that our library is useful for a general GNN practitioner. Due to the current lack of a unified API, such users often stick with default, streamlined architectures simply because it is too time-consuming to quickly try hierarchical architectures that might be more suitable, potentially losing performance along the way.
>
> Regarding the idea of making a library that shifts community practices, we believe there is no such need; PyTorch Geometric is a very well-designed and established framework, and our goal is to improve it by providing a seamless extension. We believe `tgp` is a substantial infrastructural contribution that lowers the barrier to entry for a whole class of models, aligning perfectly with ICLR's call for software library submissions (see the general Official Comment)
>
> ### Weakness 2
>
> As a paper submitted to the ICLR software track, our goal is precisely to provide systems-level innovation through consistent APIs and robust engineering rather than new methodology and theory, which is out of scope. We disagree with the assessment that integrating scattered implementations is a "moderate effort." This fragmentation, which includes inconsistent APIs and deep architectural differences, presents a significant barrier for both practitioners and researchers.
>
> Our paper clearly showcases the novel capabilities that are extremely cumbersome to realize without `tgp`. The first one is switching between sparse and dense pooling methods; our unified API is the first to handle the fundamentally different pooling layers, batching, processing, and readout requirements, making them interchangeable. Second, our caching and pre-coarsening features are novel features. These mechanisms provide orders-of-magnitude speedups, transforming expensive coarsening methods from impractical to highly efficient. Third, our modularity uniquely enables combining existing components from different poolers to create new ones. Finally,`tgp`allows for the fair, rapid model selection that was previously infeasible.
>
> These are the new, powerful capabilities that `tgp` provides beyond simple harmonization.
>
> ### Weakness 3
>
> We believe our paper already provides this head-to-head comparison. Regarding “API ergonomics”, Listing 1  is our systematic demonstration of "lines of code to swap". It shows how our unified API handles the non-trivial architectural differences between, for example, sparse and dense poolers (which require different data handling and subsequent layers), reducing the swap to a single line.
>
> Regarding runtime, our implementations do not change the core logic of the pooling operators, only the API. In particular, to implement the poolers in our library, we relied on the original code provided by the authors or by efficient implementations in PyG. Therefore, they produce the **exact same results** as the original implementations by design.
>
> The comparison for *non-trainable* methods is the core of our efficiency analysis. Table 4 clearly shows that `tgp`'s caching and pre-coarsening system is orders-of-magnitude faster, demonstrating our substantial practical value.

---

> > ### Author Response · Authors · 2025-11-19
> > **List of changes in the updated manuscript**
> >
> > We have addressed all concerns raised by the reviewer through detailed responses and revisions to the manuscript. For convenience, we would like to highlight where these specific topics are addressed in the paper:
> >
> > - **Comparison with Existing Libraries (Spektral/DGL):** The fragmentation in the field and the specific gap our library addresses are detailed in the **Introduction (Section 1)**.
> > - **Extensibility and Modularity (Creating New Poolers):** We demonstrate the modular design and its benefits in two ways:
> >     - **Code Ergonomics:** **Section 3.2** now includes a listing that shows how to combine and instantiate custom poolers with only one line of code.
> >     - **Experimental Validation:** The practical benefit of this modularity (isolating and swapping components for fair comparison) is experimentally showcased in **Section 4.5**.

---

> > ### Comment · Reviewer_4FU5 · 2025-11-25
> >
> > Thank you for the detailed response and for the additional clarifications and revisions to the manuscript.
> >
> > After reading the rebuttal, I believe my original understanding of the paper was largely accurate. I appreciate the arguments about the importance of pooling as a fragmented, underdeveloped subcomponent and I agree that tgp offers a useful and well-engineered contribution for both pooling experts and general GNN practitioners. I also acknowledge the additional emphasis on new capabilities such as unified sparse/dense pooling, caching and pre-coarsening, and modular recombination of components; these help clarify the value beyond simple code consolidation.
> >
> > That said, my main concern remains about the overall scope and impact for a long, archival paper at a major machine learning conference, even within the software track. In my view, the work is still primarily an infrastructural unification and engineering effort around a single GNN subcomponent, rather than a broadly enabling system or a qualitatively new abstraction that would significantly shift community practice. The added explanations help, but they do not fully change my assessment of the contribution level relative to the venue’s bar.
> >
> > In summary, I recognize that the library is well designed and could be genuinely helpful to practitioners, and I appreciate the improvements made in the revision. Nonetheless, my overall evaluation and score remain unchanged.

---

> > > ### Author Response · Authors · 2025-11-26
> > >
> > > Thank you again for reading and for acknowledging the usefulness and engineering quality of the library.
> > >
> > > The judgment about scope and impact inevitably involves some subjectivity. We would like to clarify how we understand the intended role of ICLR’s software track. In recent years, several libraries targeting a specific modelling building block — rather than an entire end-to-end platform — (e.g., Einops, BatteryML, etc...) have been accepted at ICLR. These works are valued not because they shift all of machine learning practice at once, but because they provide well-engineered abstractions that significantly lower the barrier to research in their specific subdomain. We believe our library fits these characteristics, enabling fast and effective building hierarchical GNNs. This has already been acknowledged by several reviewers, who note that it “addresses a real need” and is “a good contribution to the GNN community”.
> > >
> > > We therefore feel that the remaining concern is not about the technical strength or clarity of the work (on which there seems to be consensus) but about how narrowly or broadly the software libraries track's impact criterion should be interpreted. Our understanding, based on the CfP and on prior accepted software-track papers, is that impactful, domain-specific infrastructure of this kind does fall within ICLR scope.

---

### Official Review · Reviewer_CPpP · 2025-10-31

**Soundness:** 3
**Presentation:** 3
**Contribution:** 3
**Rating:** 4
**Confidence:** 3

**Summary:**

The paper introduces Torch Geometric Pool (tgp), a library built on PyTorch Geometric (PyG) that unifies various graph pooling methods under a consistent API. By using the Select-Reduce-Connect-Lift (SRC(L)) framework, tgp provides a modular design for both trainable and non-trainable, sparse and dense poolers. The work implements 17 pooling methods spanning trainable/non-trainable and sparse/dense paradigms, provides caching and precoarsening mechanisms for efficiency, and includes systematic benchmarks across unsupervised clustering, node classification, and graph-level tasks on 20+ datasets.

Overall, tgp aims to simplify experimentation, enhance the usability of graph pooling, and facilitate rapid prototyping in GNN research.

**Strengths:**

(1)Addresses a real need. The fragmentation of pooling implementations across frameworks is a genuine pain point. The paper's motivation—enabling fair comparison and rapid prototyping—is compelling and well-articulated.

(2)Comprehensive graph pooling benchmark: The paper presents a thorough benchmarking of multiple pooling operators across several tasks, highlighting the trade-offs and aiding researchers in selecting the best pooling method for their specific application.

(3)Novel efficiency mechanisms. The caching and precoarsening features are practical innovations that meaningfully advance the usability of deterministic poolers.

**Weaknesses:**

(1) Limited contribution. The SRC(L) framework itself is not a novel contribution; it is attributed to Grattarola et al. (2022). Consequently, the primary contribution of this paper lies in the engineering effort of building a comprehensive, well-designed software library around the existing framework, rather than in proposing new architectural principles or algorithmic insights for graph pooling itself.
[Reference] Grattarola, Daniele, et al. "Understanding pooling in graph neural networks." IEEE transactions on neural networks and learning systems 35.2 (2022): 2708-2718.

(2) Missing comparisons to recent unified frameworks: While the paper addresses the fragmentation of graph pooling implementations, it does not provide direct comparisons to other recent unified frameworks.

(3) Lack of concrete evidence for extensibility. The paper emphasizes SRC(L) modularity as a design strength, claiming users can "combine new and existing modules" to create novel poolers. However, no experimental ablations validate this claim. It would strengthen the paper to include practical examples or experiments that demonstrate how easy it is for users to create new poolers by combining different modules from the library.

**Questions:**

(1) The paper addresses the fragmentation of pooling implementations, but it does not directly compare tgp to other recent unified frameworks like Spektral or Deep Graph Library (DGL). Could you provide a comparison of tgp with these other libraries, especially in terms of their pooling capabilities, ease of use, and performance?

(2)Beyond engineering, does the library introduce conceptual innovations? The SRC framework is from Grattarola et al. (2022), and all 17 operators are existing methods. Does tgp propose new pooling paradigms, theoretical analysis, or novel inductive biases?

(3)In the unsupervised node clustering experiments, the paper uses default hyperparameters for each pooling method. However, as you mention, hyperparameter tuning can significantly impact the performance of different poolers. Could you provide more details on how you plan to address the tuning for each method? Specifically, how does the performance change with tuned hyperparameters, and could a controlled tuning experiment be conducted to demonstrate fair comparison across the poolers?

---

> ### Author Response · Authors · 2025-11-19
> **Rebuttal (1/2)**
>
> Thanks for your review. We address your questions and comments below.
>
> ## W1: limited contribution, engineering effort, no principles/algorithmic insights
>
> Our contribution is a software library and perfectly aligns with the topic “infrastructure, software libraries, hardware, etc.” listed in the ICLR CfP (see the general Official Comment).
>
> ## W3: Lack of concrete evidence for extensibility.
>
> We agree that extensibility is a core design goal of `tgp`. However, we respectfully believe that a quantitative ablation study is not feasible in this context, as extensibility is a qualitative architectural feature rather than a performance metric.
>
> The "concrete evidence" for this capability lies in the library's design, grounded in the standardized interfaces discussed in **Section 3.2** and **Appendix B** . The `SelectOutput` and `PoolingOutput` structures define a strict exchange protocol between the modular SRC(L) components. This design provides a structural guarantee: any new `Select` module is automatically compatible with all existing `Reduce` and `Connect` modules, provided it respects the interface contract. Since this compatibility is enforced by the API structure, it requires no experimental validation.
>
> Nevertheless, to address the reviewer's request for practical examples and concrete validation, we have significantly revised the manuscript:
>
> - We added **Listing 3 in Section 3.2**, which demonstrates how to create a custom pooler in a single line of code by mixing different existing components.
> - We introduced a new experimental evaluation in **Section 4.5**. Here, we assess `Top-k` and `k-MIS` poolers combined with two different connectivity modules (`SparseConnect` and `KronConnect`). This experiment serves as practical proof of how `tgp` allows users to isolate specific components—keeping the selector constant while swapping the connector—to measure their independent impact on performance.
> - We included a **tutorial** in the supplementary material (and documentation) that walks users through the creation of a novel pooling operator, further demonstrating the ease of prototyping.
>
> ## W2 and Q1: lack of comparison to recent unified frameworks
>
> It is not clear how such a comparison should be performed, since PyG, Spektral, and DGL serve different ecosystems and, more importantly, different purposes.
>
> - PyG has become the *de facto* standard for GNN research in the PyTorch ecosystem, and our goal was to build a seamless extension for that community. As we highlight in the introduction, most graph pooling methods are available only in some of the repositories associated with the original papers presenting the method.
> - Spektral, instead, is primarily a TensorFlow/Keras-based library and therefore in a different ecosystem, and is no longer actively maintained. Its collection of hierarchical poolers is also not as comprehensive as `tgp`'s (8 vs 17 poolers).
> - DGL, on the other hand, only includes global pooling operators, which are analogous to the standardized "Aggregators" already available in the core PyG library (which is why we did not re-implement them). `tgp`'s focus is on hierarchical pooling (graph-to-graph operations), a much more complex and fragmented area for which DGL does not provide a comprehensive or unified framework.
>
> We clarified these points better in the introduction of the revised manuscript.
>
> ## Q2: conceptual innovations beyond engineering
>
> We do not introduce new pooling paradigms, theoretical analysis, or novel inductive biases because it is out of scope for this work.
>
> Beyond the engineering of consolidation, `tgp` introduces some key conceptual innovations. The most significant is our caching and pre-coarsening system, which makes the implementation of the pooling operators much more efficient than existing ones. This is a novel mechanism that decouples static graph coarsening from feature-dependent operations. Implementing this efficiently required the design of new, standardized data structures, such as the `PooledBatch`, which manages and collates this pre-computed information during training.
>
> A second core conceptual abstraction is the unified `SelectOutput` object. This standardized data structure acts as a common interface for all pooling methods, decoupling the `Select` logic from the subsequent `Reduce` and `Connect` stages. This is the key abstraction that enables our library's modularity and makes fundamentally different pooling paradigms truly interchangeable.
>
> The new experiment in **Section 4.5** of the revised manuscript showcases how `tgp` allows for the interchangeability of the pooler’s components.

---

> > ### Author Response · Authors · 2025-11-19
> > **Rebuttal (2/2)**
> >
> > ## Q3: hyperparameters in the clustering experiment
> >
> > In our experiments, we used the default hyperparameters and we commented that this can significantly impact unsupervised tasks where the auxiliary loss is the only learning signal. The issue here is that a fair tuning process for unsupervised clustering is not trivial, as there is no straightforward validation signal in the absence of supervised information. We further stressed this point in **Section 4.1**.
> >
> > We would also like to comment that our goal is not to find the state-of-the-art performance for each pooler, but to demonstrate that `tgp`'s unified API *enables* such diverse comparisons in the first place. Indeed, the modularity of `tgp` makes it an ideal tool for researchers to easily swap operators and conduct the very kind of custom grid searches the reviewer suggests, even if such an exhaustive sweep (especially in clustering, where the "optimal" partition is not well-defined) is outside the scope of this library-focused paper.
> >
> > We have clarified this distinction in the revised version.

---

> > ### Comment · Reviewer_CPpP · 2025-11-27
> >
> > Thank you for the detailed response. I acknowledge that the work aligns with ICLR's stated scope for infrastructure contributions.
> >
> > However, the core limitation remains: while the tgp provides genuine value to practitioners—unified API, efficient implementation, comprehensive benchmarking—it is fundamentally a consolidation of existing methods and frameworks. This is precisely what limits the contribution for a top-tier venue like ICLR.
> >
> > The work would be an excellent fit for a software or tools track at a specialized venue. For ICLR, the bar for infrastructure papers requires either substantial algorithmic novelty within the infrastructure or exceptionally broad impact. While this library is well-designed and useful, it does not quite meet that threshold.
> >
> > I will maintain my original score. The paper represents quality engineering work, but falls marginally below the acceptance bar for this venue.

---

> > > ### Author Response · Authors · 2025-12-01
> > >
> > > We appreciate that you acknowledge both that the work "aligns with ICLR’s stated scope for infrastructure contributions" and that `tgp` provides "genuine value to practitioners".
> > >
> > > We would like to note that the ICLR CfP for infrastructure / software libraries does **not** state that such papers must contain substantial new algorithms or have "exceptionally broad impact"; rather, they are expected to enable and accelerate research. Recent ICLR software-track papers similarly systematize and unify existing techniques within a focused domain.
> > >
> > > Within hierarchical GNNs, `tgp` plays an analogous role. We therefore believe that, given the CfP and these precedents, our work satisfies the intended "bar" for an infrastructure contribution, even if the level of such a "bar" remains necessarily subjective.

---

### Author Response · Authors · 2025-11-14
**Software libraries track on ICLR**

We thank the reviewers. A common theme was the concern that a software library paper, which by nature focuses on engineering, infrastructure, and usability, might be a better fit for a different venue than ICLR.

We believe this view stems from a misunderstanding of ICLR's thematic areas. The ICLR Call for Papers has, for several years, included a track for **"infrastructure, software libraries, hardware, etc."**. The value of such contributions lies not in novel theory, but in providing the community with well-designed and efficient tools that enable and accelerate new research. Our library, `tgp`, aligns perfectly with this call, as it moves hierarchical GNNs from being a complex, ad-hoc research topic to an easily accessible tool for any practitioner.

ICLR has a strong and recent history of publishing high-impact papers on "handy, useful packages" that are valued for their infrastructural contributions. Examples of such papers accepted at ICLR include:

- **Einops: Clear and Reliable Tensor Manipulations with Einstein-like Notation** (Rogozhnikov, ICLR 2022, https://openreview.net/forum?id=oapKSVM2bcj)
- **TorchRL: A data-driven decision-making library for PyTorch** (Bou et al., ICLR 2024, https://openreview.net/forum?id=QxItoEAVMb)
- **BatteryML: An Open-source Platform for Machine Learning on Battery Degradation** (Zhang et al., ICLR 2024, https://openreview.net/forum?id=sxGugrYhP9)
- **CAX: Cellular Automata Accelerated in JAX** (Faldor & Cully, ICLR 2025, https://openreview.net/forum?id=o2Igqm95SJ)
- **Betty: An Automatic Differentiation Library for Multilevel Optimization** (Choe et al., ICLR 2023, https://openreview.net/forum?id=LV_MeMS38Q9)

We kindly ask the reviewers to familiarize themselves with this track and consider our work among these contributions.

---

### Author Response · Authors · 2025-12-02
**Summary for the AC**

Dear AC,

We would like to provide a summary of our rebuttal. We **addressed every question** raised by the reviewers and **significantly revised the manuscript** to include new content and experiments.

Regarding the technical content, all reviewers were satisfied after the rebuttal, except for Reviewer osFv, who expressed additional concerns. However, in our last answer, we argued that these concerns were **grounded on erroneous understanding** regarding basic functionalities in PyG and the architecture of established baselines.

The main concern that remains is whether ICLR is a good fit for `tgp`.

All reviewers explicitly acknowledged that the library is "well-designed," "genuinely helpful," "Addresses a real need", represents "quality engineering", is "A good contribution to the GNN community", and "aligns with ICLR's stated scope for infrastructure contributions". However, in their opinion, a software library focusing on "consolidation" or "engineering" does not meet the "impact" bar for ICLR because it lacks algorithmic novelty.

We would like to highlight the following points:

- Requirements of algorithmic novelty for a software library are **not** part of the ICLR CfP, which explicitly invites submissions on **"infrastructure, software libraries, hardware, etc."**
- We believe the reviewers are applying **criteria suited for methodology papers** (requiring novel theory or algorithms) to the software track, which rather prioritizes usability, unification, and accelerating research.
- As noted in our general comment, ICLR has a strong **history of accepting contributions like ours** (e.g., Einops, TorchRL, BatteryML, CAX, Betty).
- There were previous cases (see e.g., https://openreview.net/forum?id=oapKSVM2bcj) where reviewers were skeptical about publishing a software paper to ICLR, but **the AC decided to accept the paper**, recognizing the usefulness of the tools for the community.
- **In this current ICLR edition, there are other submissions on software libraries** (see e.g., https://openreview.net/forum?id=kFgsebdKje) that are appreciated for their utility and engineering quality, suggesting that `tgp` is a good fit for this venue.

Thank you for your time and consideration,

the Authors.

---

### Meta-Review · Area_Chair_HtWm · 2026-01-06

**Summary:**

This paper introduces TGP, a library for hierarchical pooling in graph neural networks. TGP unifies a variety of pooling operators under a common Select–Reduce–Connect–Lift framework. Experiments on several downstream tasks highlight the potential value of a flexible library for rapid prototyping and evaluation. However, the reviewers raised several concerns, including the lack of comparisons with existing frameworks such as Spektral or DGL, insufficient empirical evaluation, and potential scalability issues related to the in-memory caching design. Thus, I recommend rejecting this paper.

**Reviewer Concerns:**

The rebuttal addresses some minor clarification points regarding the design of the TGP framework. However, the main concerns raised by the reviewers remain largely outstanding. In particular, the rebuttal does not provide comparisons with other established frameworks such as Spektral or DGL, does not substantially strengthen the empirical evaluation, and does not adequately address the scalability concerns associated with the in-memory caching mechanism.

**Reviewer Scores:**

The rebuttal does not resolve concerns regarding in-memory caching and scalability. As a result, the reviewer’s overall assessment and score would likely remain unchanged.

---

### Decision · Program_Chairs · 2026-01-26

Reject